# Circulating Osteogenic Progenitor Cells Enhanced with Teriparatide or Denosumab Treatment

**DOI:** 10.3390/jcm11164749

**Published:** 2022-08-14

**Authors:** Mercè Giner, María Angeles Vázquez-Gámez, María José Miranda, Jesús Bocio-Nuñez, Francisco Jesús Olmo-Montes, Miguel Angel Rico, Miguel Angel Colmenero, María-José Montoya-García

**Affiliations:** 1Bone Metabolism Unit, UGC Medicina Interna, Hospital Universitario Virgen Macarena, Avda. Dr. Fedriani s/n, 41009 Sevilla, Spain; 2Departamento de Citología e Histología Normal y Patológica, Universidad de Sevilla, 41009 Sevilla, Spain; 3Departamento de Medicina, Universidad de Sevilla, Avda. Dr. Fedriani s/n, 41009 Sevilla, Spain

**Keywords:** circulating osteogenic progenitor cells, osteoporosis, metabolic bone disorders, Teriparatide, Denosumab

## Abstract

Circulating osteogenic precursor (COP) cells are peripheral blood cells with a capacity for osteogenesis. The objective of our study was to ascertain the percentage of COPs as an early biomarker of osteoporosis and the effect of these cells in response to Denosumab (DmAb) (anti-resorptive) or to Teriparatide (TPDP) (anabolic) as very effective drugs in the treatment of the illness. A first study was conducted on healthy volunteers, with three age ranges, to determine the percentage of COPs and relate it to their anthropometric and biochemical characteristics, followed by a second longitudinal study on patients with osteoporosis, whereby one group of patients was treated with TPTD and another with DmAb. All were analyzed by cytometry for COP percentage in blood, bone turnover markers, and bone mass. Our findings show that COPs are influenced by age and become more prolific in the stages of growth and skeletal maturation. A higher percentage of COPs is found in osteoporotic disease, which could constitute a predictive marker thereof. We also show how treatment with TPTD or DmAb mobilizes circulating osteogenic precursors in the blood. Significant increases in % COPs were observed after 12 months of treatment with Dmb (21.9%) and TPTD (17%). These results can be related to an increase in osteogenesis and, consequently, a better and more efficient repair of bone tissue.

## 1. Introduction

Circulating osteogenic precursor (COP) cells are peripheral blood cells with the capacity for osteogenesis [1]. Despite there being controversy regarding their origin, one of the most accepted ideas is that they come, at least in part, from bone marrow, from mesenchymal cells (MSC), and that they present the ability to circulate on demand for the needs of the bone peripheral tissue [2,3,4]. These COPs are estimated as representing approximately 0.42% of peripheral blood mononuclear cells (PBMC), but they present a very wide % range (0.1–34%) that depends on age; gender; pathological factors [5,6]; and certain peripheral stimuli, such as substance P [7] and hypoxia [8]. In this subpopulation, it has also been possible to verify bone-formation cell markers such as osteocalcin (OCN), type 1 collagen (COL1), and alkaline phosphatase (AP).

Khosla et al. have identified and isolated COPs in human peripheral blood by means of flow cytometry, using bone-formation markers such as OCN. Osteocalcin is a soluble protein that is produced by COP, and it can remain anchored in the plasma membrane due to its Gla residues; this characteristic enables the identification of COPs (OCN+) circulating in blood [5]. It has consequently been shown that COPs (OCN+) are mobilized both in situations of fractures that occur during puberty and in primary osteoporosis (OP) [9], whereas they are reduced in various pathological processes, such as in secondary forms of induced osteoporosis (e.g., due to glucocorticoids, in Type 2 Diabetes Mellitus (DM2) [3,10,11,12]), and in situations of greater disability and frailty [13].

In OP, there is an imbalance between bone formation and resorption that leads to loss of bone mass. Today, for the treatment of the disease, anti-resorptive drugs such as Denosumab (DmAb) and anabolic drugs such as Teriparatide (TPTD) are available. In both cases, the main objective is to prevent bone loss and to increase bone formation, either by slowing resorption or boosting formation. In the case of the anti-resorptive drug, DmAb is an anti-RANKL IgG2 monoclonal antibody that prevents the coupling of RANKL with its RANK receptor, which is responsible for the activation of the nuclear factor NF-B in cells of osteoclastic lineage. As a consequence, it inhibits the formation, function, and survival of osteoclasts, thereby inhibiting bone resorption. Treatment with DmAb for 3 years reduces the incidence of vertebral fractures (the relative risk reduction being 61% in the first year, 78% in the second, and 65% in the third) and hip fractures (with a 40% relative risk reduction). It has also been shown that, subsequent to DmAb treatment, bone mineral density (BMD) increases significantly in the lumbar spine by 9.2% and in the total hip by 6.0%, as compared to using a placebo (*p* < 0.001) [14]. Moreover, regarding the anti-resorptive effect, this could be a pro-osteogenic effect that justifies the change in BMD [15]. Its favorable skeletal effects reverse quickly upon its discontinuation due to a vast increase in the osteoclast number and activity, which leads to a subsequent profound increase of bone turnover above pretreatment values, a phenomenon commonly described as the “rebound phenomenon” [16].

On the other hand, Teriparatide (a recombinant parathyroid hormone (1–34)) is an anabolic drug that induces de novo bone formation by increasing the rate of bone remodeling in favor of formation with increased trabecular connectivity and cortical bone thickness [17]. Its administration induces bone resorption by activating osteoclasts indirectly through their actions on osteoblastic cells and inhibits the sclerostin production by the osteocyte [18]. Subsequent to two years of treatment, significant reductions are observed in the rate of new vertebral (65%) and non-vertebral (53%) fractures. At the BMD level, it produces a net increase of 9.7% in the lumbar spine and 2.8% in the femoral neck [19].

In osteoporotic disease, given the timeline required to verify the positive effects of drugs on bone mass and fracture-risk reduction, serum determinations of bone turnover markers (BTMs) have been proposed at basal and after 3 months. The BTMs proposed are C-terminal telopeptide of type 1 procollagen (CTX) as a marker of reabsorption, and of N-terminal propeptide of type 1 procollagen (P1NP) of formation, in order to predict future efficacy of the treatment [20].

In the few studies available, significant data show the relevance of COP cells in the pathogenic mechanism of osteoporosis. The effect of some anabolic and anti-resorptive drugs on these cells has been assessed, but the effect of DmAb remains unknown.

Our study objectives therefore include the determination of the percentage of COP(OCN+) in healthy volunteers of different ages and in patients with primary osteoporosis, as well as the evaluation of the variation of the percentage of COP(OCN+) in peripheral blood after anti-resorptive treatment or anabolic treatment.

## 2. Materials and Methods

### 2.1. Study Design and Subjects

A first study was previously conducted on healthy volunteers, across three age ranges, to determine the percentage of COP and relate it to their anthropometric and biochemical characteristics. No pathologies related to bone metabolism, such as osteoporosis, were presented by the subjects included. These subjects (*n* = 35) were evaluated as three control groups: (a) 13 children (9–11 years old), (b) 13 young adults (26–32 years old), and (c) 9 senior adults (>60 years old). The female/male ratio remained close to 1 for the whole analysis.

For the investigation, a second longitudinal study was conducted in a single center from February 2019 to February 2020 on patients with osteoporosis. In order to qualify for inclusion in the study, the patients had to be aged > 55 and diagnosed with BMD < −2.5, without a recent history of any fragility fracture and without having received any previous osteoporotic treatment. One group of patients was treated with Teriparatide (*n* = 7) and another with Denosumab (*n* = 10) for one year. All patients were women. The patients were excluded from participating if they were receiving any drug that might affect bones, such as glucocorticoids, glitazones, immunosuppressants, anti-resorptive therapies, drugs that stimulate bone metabolism formation, thyroxine, anti-convulsants, and anti-coagulants, or if they exhibited an endocrine or systemic disease that might affect bone metabolism. We required there to be 5 subjects in each group in order to detect a significant standardized mean difference of 0.5 (one size average effect) with a type 1 error rate of 5% (alpha = 0.05) and a 90% power (1-beta = 0.90). The number of patients remained the same throughout the longitudinal study except in the 12-month analysis, where the number of patients treated with TPTD was 6 instead of 7, due to the abandonment of treatment.

The study was approved by the Ethics Committee of the Virgen Macarena Hospital (Sevilla, Spain). Before applying any study procedure, written informed consent was obtained from every subject.

Furthermore, a designed protocol was followed by all patients (Figure 1). A fasting blood sample was obtained from every subject for study as BTM, biochemical determinations, and a cytometric analysis of COP cells at basal and at 3, 6, and 12 months after treatment. BMD was also measured at basal and 12 months after treatment.

### 2.2. Biochemical Measurements

Fasting morning blood was drawn, and serum was stored at −80 °C. CTX, P1NP, and the insulin growth factor (IGF-1) were analyzed in an immunoassay by an autoanalyzer COBAS 601 (Roche, Spain): inter-assay coefficient of variation (CV) <5.8%, <7.6%, <4.2%, and <5.1%, respectively.

Calcium and phosphorus were analyzed by an autoanalyzer (ADVIA 2400, Siemens): CV intra-assay, 0.8%; and inter-assay, 1.5%.

Then 25 hydroxyvitamin D3 (25OHVD3) was analyzed in a chemiluminescence immunoassay by using an autoanalyzer (LIASON, DiaSorin, Italy): CV intra-assay <5.5% and inter-assay <3.5%.

### 2.3. Isolation of Peripheral Blood Mononuclear Cells (PBMCs)

Ten milliliters of whole blood was diluted by vacuum venipuncture in hematized HD tubes with 10 mL of PBS (1:1). The diluted blood was distributed with a Pasteur pipette in conical-bottom test tubes in a Ficoll-Hypaque 1:2 solution and centrifuged (1250× *g*, 20 min, 4 °C). The ring that formed at the interface contained the mononuclear cells that were then collected and washed with PBS. A hypo-osmotic shock was subsequently performed with distilled and deionized water to lyse the contaminating red blood cells; 1.8% NaCl was added to restore osmolarity. It was then washed with PBS, the cells were counted with a hemocytometer, and the viability was determined by using the trypan blue test.

### 2.4. Cytometry Analysis of COPs

The cells were incubated with 10% donkey serum (Jackson ImmunoResearch, West Grove, PA, USA) and 10% human IgG (FcR Blocking Reagent, Milteny Biotec, Bergisch Gladbach, Germany) for 30 min at room temperature, under agitation, to block any non-specific binding sites. Aliquots of 106 cells were then transferred into 5 mL round polystyrene tubes and incubated for 60 min at room temperature with primary human anti-osteocalcin antibodies (R&D Systems, Minneapolis, MIN, USA). The cells were washed twice with PBS (centrifuged at 4 °C at 400× *g* for 2 min) to remove free antibodies and then incubated with a secondary antibody, GoatF(ab’)2 fragment anti-mouse IgG (H + L)-FITC (Beckman Coulter, Brea, CA, USA) at room temperature for 30 min in the dark. At the same time, two tubes were incubated containing the same cell density but only containing the secondary antibody as an isotype control to measure non-specific staining. All incubations were performed twice; after the second incubation, the cells were washed twice with PBS, centrifuged at 4 °C at 400× *g* for 2 min, and then transferred on ice to the flow cytometer [21].

### 2.5. Bone Mineral Density

The BMD of the lumbar spine (L1-L4), femoral neck, and total hip was measured by dual-energy X-ray absorptiometry, using the Hologic Discovery W densitometer (Hologic, Inc, Waltham, MA, USA) equipped with APEX 3.1.1 software. The in vivo CVs were 2.4% (femoral neck), 1.1% (total hip), and 3.8% (lumbar spine L1–L4).

### 2.6. Statistical Analysis

For descriptive analyses, continuous data are presented as the means and standard deviation, and categorical data are presented as absolute and relative frequency. For BMD, age-adjusted T scores were calculated. Normality was tested with the Shapiro–Wilk test. Differences between groups were analyzed by using one-way analysis of variance (ANOVA) for multiple comparisons and Student’s *t*-test. Correlations between continuous outcomes were examined by using the Spearman or Pearson correlation coefficient.

All statistical analyses were performed by using the Statistical Package for Social Sciences (SPSS) v.23.0 (IBM Corp., Armonk, NY, USA). All tests were two-sided and considered significant if *p* < 0.05.

## 3. Results

### 3.1. Basal Anthropometric Parameters, BTMs, and BMD in the Healthy Population

The characteristics of the healthy volunteers are shown in Table 1. There were no gender differences between the study groups. Related to BMI, significant differences were observed in that the children had a lower BMI; hence, the studies were adjusted for BMI, and no differences in the subsequent statistical analysis were encountered. The group of children showed the highest levels of P1NP, with statistically significant differences versus the young adults (*p* = 0.0001) and the senior adults (*p* = 0.0001). The remaining parameters were similar, although the group of children showed an upward trend in CTX concentration. The BMD values lie within the expected range for their age and gender.

### 3.2. Percentage of COP Cells in Healthy Population

The COP (OCN+) levels were measured in each group. A decrease in relation to age was identified, whereby the senior group showed less %COP than the groups of healthy young adults (*p* = 0.004) and children (*p* = 0.0001). Significant differences were also found between the groups of children and young adults (*p* = 0.034) (Figure 2). The percentage of COP shows a positive correlation with the formation marker P1NP (r = 0.536; *p* = 0.002).

### 3.3. Basal Anthropometric Parameters and BTMs and BMD in OP Patients

The baseline characteristics are shown in Table 2. The average age of the TPTD treatment group was similar to that of the DmAb treatment group, as were the remaining parameters. We found no differences in BMI or BTM values. The OH-vitamin D3 levels stayed within the normal range.

### 3.4. Percentage of COP Cells in Osteoporosis Patients

The subjects of the treatment study present osteoporotic disease. When analyzing the basal % of COPs among patients with osteoporosis disease and those among the senior group, we observed that OP patients present a higher percentage, 8.4 ± 7.03 vs. 3.5 ± 1.94 (*p* = 0.027). No correlations with BMD or BTM values were observed.

### 3.5. Percentage of COPs Correlating with the Treatment

During the study, an increase in the percentage of circulating COPs in both groups was observed 12 months after treatment (Figure 3). Specifically, the TPTD group underwent an early increase in their percentage of COP; at 3 months from the start of treatment, a significant increase of almost double the COP cells as compared to the baseline situation had already been observed (*p* = 0.003), and from 6 months, the values of COP were maintained (*p* = 0.044). In contrast, the DmAb group showed a slight early decrease during the first 3 months of the study, although a continuous increase of COP was observed until the end of the experiment. Taking all of this into account, after 12 months of treatment, the final percentage of COP was similar for the two groups: 19.91 ± 3.05% in the TPTD group and 18.08 ± 3.84% in Dmab. However, a statistically significant difference was observed between the basal condition and after 12 months of treatment with DmAb (*p* = 0.004).

### 3.6. BTMs after 12 Months of Treatment and Correlation with COP

During DmAb treatment, a decline followed by steady P1NP levels were observed. The decrease between baseline and 3 months, after receiving treatment, was 42% (*p* = 0.07) and the decrease was maintained at 6 months (*p* = 0.008) and at 12 months (*p* = 0.017) (Figure 4). In the TPTD group, we found no differences.

The CTX levels were higher in the TPTD group than in the DmAb group and both had a similar pattern (Figure 3).

Vitamin D levels remained within normal ranges throughout the treatment and in both groups. Furthermore, IGF-1 values remained constant.

In correlation studies, we observed that COP(OCN+) levels correlate with P1NP levels (*r* = 0.702; *p* = 0.008) in the TPTD group.

### 3.7. BMD

Studies were carried out on the femoral neck, hip, and lumbar spine BMD (gHA/cm^2^) at the basal and 12 months after treatment (Table 3 and Table 4). A slight increase was observed in femoral neck after treatment with TPTD (3.1%), and in the group treated with DmAb, an increase of 5.7% was observed in lumbar spine at 12 months. The remaining locations presented no differences after 12 months of treatment.

## 4. Discussion

With the present study, it is verified that the percentage of COPs can be quantified in circulating blood and that this percentage is higher during the ages of development and skeletal maturation when compared to older ages, as is in accordance with the indications of other authors [3,4,22,23,24,25]. These results lead us to consider future studies in which verification is made of the usefulness of quantifying these cells in the early stages of diseases in which bone metabolism may be compromised.

It is also shown that, in primary osteoporosis, there is an increase in bone remodeling that produces a net balance of increased osteoclastogenesis. Certain authors consider that the body tries to modulate this balance by increasing circulating COPs. In our case, in basal conditions, a higher % of COP was observed in the PO group than in the same group of the same age without OP [15,26,27,28].

When studying the effect of treatments with TPTD or DmAb, we observed that, in both cases, at 12 months, there was an increase in COP (OCN+), and this may indicate an increase in the activity of the bone remodeling process by recruiting a greater number of osteoprogenitors throughout the treatment.

As expected, the changes in the percentage of COP present in the blood after treatment with Teriparatide are very prompt. An increase at 3 months is observed, and this is maintained throughout the study. The presence of these osteoprogenitors in the tissue is stimulated by the drug, thereby activating bone formation, and this falls in line with the increased values of P1NP and CTX, thus indicating a strong association between the two coupled processes in bone remodeling. These results coincide with those of other authors [10,29,30] and suggest that Teriparatide can mobilize and recruit progenitors with osteogenic capacity that contribute toward bone regeneration, thereby increasing osteogenesis, and toward the expected increase in BMD in these patients.

Patients treated with Denosumab, a potent anti-resorptive agent, present BTM values in accordance with the expected pattern. These patients present a clear and rapid decrease in CTX, whereby it becomes almost undetectable in the first days of administration. P1NP is also reduced, but not as markedly and at a slower pace. COP(OCN+) values in these patients begin to increase after 3 months when P1NP levels have already reached their minimum value, and this implies that there is a coupling between the process of resorption and bone formation in these patients from 3 months and that it is maintained during the 12 months of study. These data also suggest an increase in the recruitment of osteogenic precursors, as with anabolic treatment, and this could partly explain the significant and progressive increase in bone mass that is described with this treatment after 12 months in the lumbar spine; this also coincides with our BMD results at that location [31,32,33]. Other authors have shown similar results with other anti-resorptive drugs such as alendronate [9] and zoledronic acid [15]. Further analyses with longer treatment times are necessary to confirm these results.

After treatment, patients with TPTD present a slightly higher % of COPs than those with DmAb, although patients with DmAb yield a better % change with respect to the baseline. It is difficult to determine the effects in each group.

The study suffers from various limitations, such as the small number of patients studied and the shortness of the study period, with only one year of treatment. However, the study population has been highly homogeneous; therefore, fewer subjects were needed to see an effect. On the other hand, our results demonstrate an increase in the recruitment of COPs, although we cannot prove that there is an increase in the maturation and differentiation of these cells that directly demonstrates their involvement in the increase in bone formation that occurs in these treated patients.

Circulating osteogenic precursors can constitute potential candidates for biomarkers in osteoporotic disease since they can indicate the therapeutic efficacy of treatment and the adherence to such treatment: the maintained upward percentage of COPs (OCN+) indicates correct and suitable treatment. On the other hand, these precursors can become potential targets for direct regenerative treatment. Advances in phenotypic and genetic knowledge of COPs can enable us to become the protagonists in cell therapy for the treatment of OP and other diseases of bone metabolism.

## 5. Conclusions

Our findings show that COPs are influenced by age, in that they are higher in the stages of growth and skeletal maturation. Not only did we show that treatment with TPTD produces an early and sustained increase in COPs, but, for the first time, we also showed how treatment with DmAb mobilizes osteogenic precursors in the blood. These results can be related to an increase in osteogenesis and, consequently, present a better and more efficient repair of bone tissue. Furthermore, the percentage of COPs in the blood can be employed in personalized medicine to determine the efficiency and/or adherence to treatment.

## Figures and Tables

**Figure 1 jcm-11-04749-f001:**
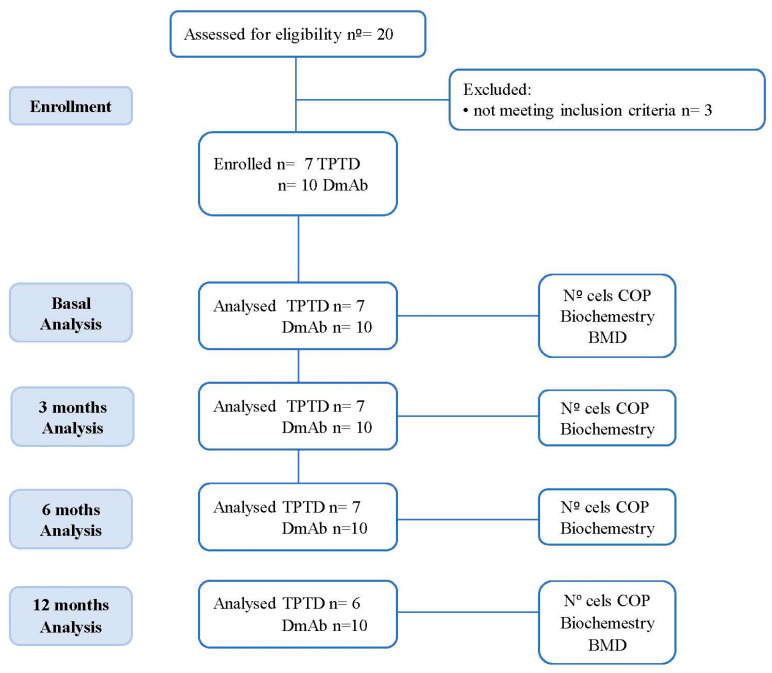
Flow diagram of the longitudinal study design. TPTD = Teriparatide; DmAb = Denosumab; COP = Circulating osteogenic precursor; BQ = Bioquimical determinations; BTM = Bone turnover markers; BMD = Bone mineral density.

**Figure 2 jcm-11-04749-f002:**
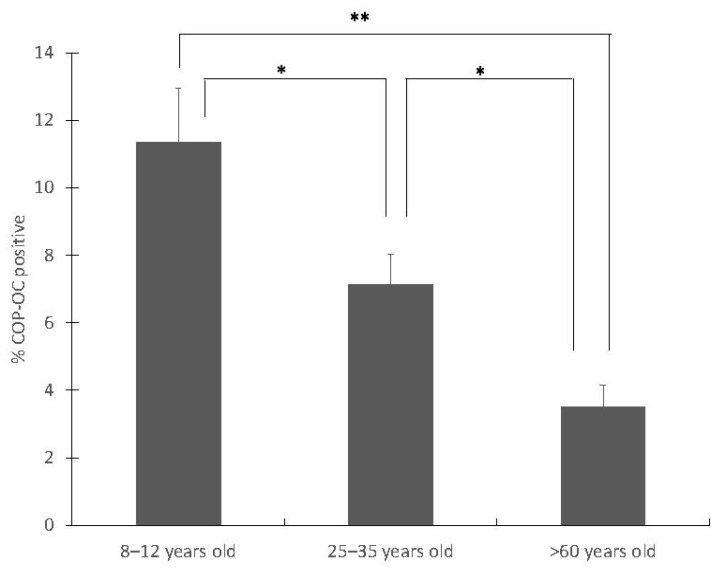
Percentage of COP (OCN+) among the control groups defined (children, *n* = 13; young adults, *n* = 13; and senior adults, *n* = 9; * *p* < 0.05; ** *p* < 0.001).

**Figure 3 jcm-11-04749-f003:**
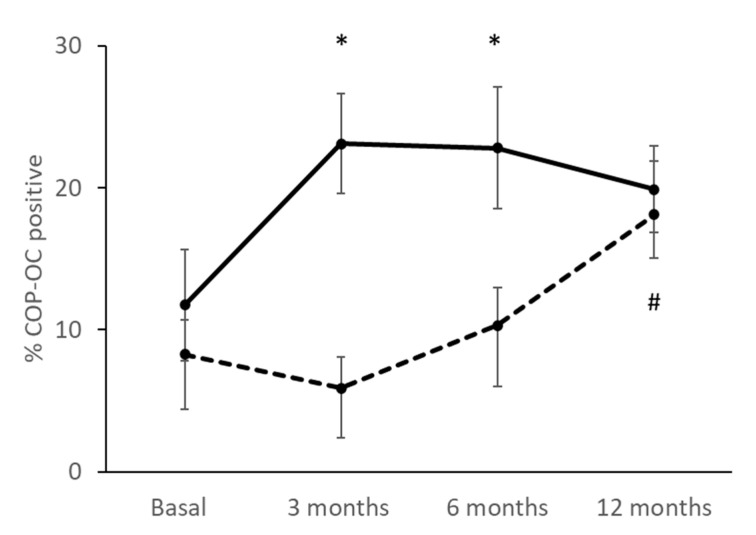
Tracking of % COP (OCN+) in each treatment group: Teriparatide (TPTD, solid line; *n* = 7) (basal vs. 3 months and basal vs. 6 months * *p* < 0.05) and Denosumab (DmAb, dashed line; *n* = 10) (basal vs. 12 months # *p* < 0.05).

**Figure 4 jcm-11-04749-f004:**
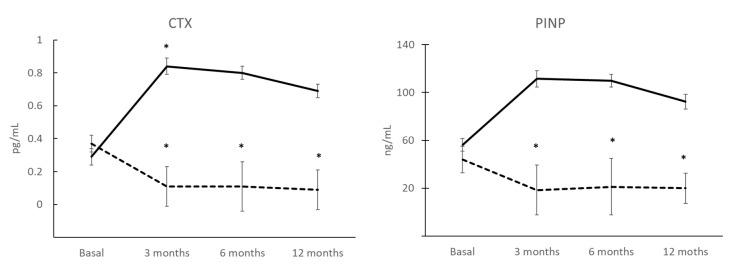
BTMs levels (CTX and P1NP) during TPTD (solid line; *n* = 7) and DmAb (dashed line; *n* = 10) treatment (basal vs. treatment: * *p* < 0.05).

**Table 1 jcm-11-04749-t001:** Baseline demographic and clinical characteristics.

	Children	Young Adults	Senior Adults
	*n* = 13	*n* = 13	*n* = 9
**Age (years)**	10 ± 1	29 ± 3	68 ± 8
**Gender (female/male)**	6/7	7/6	6/3
**Body mass index (kg/m^2^)**	20.41 ± 4.5 *	22.80 ± 2.3 *	25.9 ± 3.3
**P1NP (ng/mL)**	485.9 ± 154.1 *	42.9 ± 3.7	41.5 ± 13.9
**CTX (pg/mL)**	1.22 ± 0.103	0.36 ± 0.04	0.31 ± 0.03
**25OHVD3 (ng/mL)**	24.06 ± 7.4	23.7 ± 1.6	32.44 ± 5.9
**Ca (mg/dL)**	9.8 ±0.1	9.4 ± 0.1	9.6 ± 0.1
**P (mg/dL)**	4.5 ± 0.1	3.4 ± 0.1	3.5 ± 0.1
**Femoral neck (gHA/cm^2^)**	0.723 ± 0.92	0.828 ± 0.14	0.622 ± 0.06
**Hip (gHA/cm^2^)**	0.813 ± 0.09	0.8612 ± 0.31	0.897 ± 0.12
**Lumbar spine (gHA/cm^2^)**	0.673 ± 0.09	0.969 ± 0.72	0.805 ± 0.15

All data lie within the mean ± standard deviation; * *p* < 0.05. P1NP = N-terminal propeptide of type 1 procollagen; CTX = C-terminal telopeptide of type 1 collagen; 25OHVD3 = 25 hydroxyvitamin D3; Ca = calcium; *p* = phosphorus.

**Table 2 jcm-11-04749-t002:** Anthropometric and biochemical characteristics of two treatment groups in basal conditions: Teriparatide group (TPTD) and Denosumab group (DmAb).

	TPTD (*n* = 7)	DmAb (*n* = 10)	*p*
Age (years)	58 ± 2	66 ± 10	0.142
Body mass index (kg/m^2^)	24.5 ± 3.7	26.3 ± 5.4	0.574
CTX (pg/mL)	0.28 ± 0.05	0.37 ± 0.05	0.302
P1NP (ng/mL)	56.4 ± 11.1	44.01 ± 5.3	0.274
IGF-1 (ng/mL)	158.2 ± 30.1	127.7 ± 18.2	0.393
25OHVD3 (ng/mL)	34.1 ± 7.9	35.5 ± 10.7	0.924

All data are given as the mean ± standard deviation. P1NP = N-terminal propeptide of type 1 procollagen; CTX = C-terminal telopeptide of type 1 collagen; IGF-1 = insulin growth factor 1; 25OHVD3 = 25 hydroxyvitamin D3.

**Table 3 jcm-11-04749-t003:** Basal bone mineral density and 12 months after treatment with TPTD.

	TPTD (*n* = 7)			*p*
	Basal	12 Months	% Change	
**Femoral neck (gHA/cm^2^)**	0.64 ± 0.12	0.66 ± 0.17	3.1	0.696
**Hip (gHA/cm^2^)**	0.82 ± 0.09	0.82 ± 0.11	0	0.321
**Lumbar spine (gHA/cm^2^)**	0.66 ± 0.07	0.65 ± 0.05	0	0.782

gHA = grams of Hydroxyapatite.

**Table 4 jcm-11-04749-t004:** Basal bone mineral density and 12 months after treatment with DmAb.

	DmAb (*n* = 10)			*p*
	Basal	12 Months	% Change	
**Femoral neck (gHA/cm^2^)**	0.60 ± 0.07	0.61 ± 0.07	1.6	0.126
**Hip (gHA/cm^2^)**	0.83 ± 0.11	0.82 ± 0.17	0	0.515
**Lumbar spine (gHA/cm^2^)**	0.70 ± 0.12	0.74 ± 0.12	5.7	0.363

gHA = grams of Hydroxyapatite.

## Data Availability

Data supporting the reported results are available from the authors on request.

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
