# Peer review of "Circulating Osteogenic Progenitor Cells Enhanced with Teriparatide or Denosumab Treatment"

_jcm, 2022, doi:10.3390/jcm11164749_

Round 1

Reviewer 1 Report

The study is focused on the Circulating osteogenic precursor cells s with the capacity for osteogenesis. This is a very interesting topic to search possibilities to treat osteoporotic patients and to prepare new treatment standards.  All data require for publication is described correctly. Methodology is typical. Paper is well written with the clear figures. Text is clear and easy to read. Limitations included. As the reviewer I see that not all interesting data from your study is described in the conclusions.  Please rewrite conclusions. As this is preliminary study add control group in the future publications.

Reviewer 2 Report

The manuscript by Mercè Giner et al claims that Circulating osteogenic precursor (COP) cells are influenced by age and become more prolific in the stages of growth and skeletal maturation. A higher percentage of COPs is found in osteoporotic disease, which could constitute a predictive marker thereof. The authors also show how treatment with TPTD or DmAb mobilises circulating osteogenic precursors in the blood. However, the manuscript could be improved if the following comments and questions are addressed.

1. My main concern is related to the number of volunteers, especially for senior adults group which just have 9 volunteers (but why Gender (female/male)= 5/3 in Table 1?). The Mat & Met and figure legends section should be more precise and detailed (n). The limited number of volunteers may not be sufficient to represent a population of patients significantly. To make sure the accurate conclusion, The number of clinical volunteers is at least 20 (Gender (female/male)= 10/10) in every group.

2. The factor of Gender is an important element for osteoporosis. I suggested that female or male volunteers could be analyzed, respectively in figures and tables.

Reviewer 3 Report

The authors investigated and compared the percentage of circulating osteogenic precursor (COP) in healthy volunteers and osteoporotic patients. They also determined and compared the variation in COP percentage after 12 months treatment with Denosumab or Teriparatide. They found that younger healthy subjects and patients with osteoporosis have higher COP percentage than older healthy subjects. COP percentage significantly increased after 3 months of Teriparatide therapy, while it slightly decreased in Denosumab group. In both groups COP percentage after 12 months of therapy was similar and not significantly different as compared to baseline. The authors conclude that these results can be related to an increase in osteogenesis and consequently a better and more efficient repair of bone tissue.

This paper provides interesting evidences, although I have some remarks with particular reference to the presentation of the manuscript.

I would also recommend professional English proofreading

ABSTRACT: You generally mention that “We also show how treatment with TPTD or DmAb mobilises circulating osteogenic precursors in the blood”. As the evaluation of the variation of the percentage of COP after anti-resorptive treatment or anabolic treatment is one of your study aims, you should include some insight of the results obtained in the abstract, as you did for the determination of the percentage of COP in healthy volunteers of different ages and in patients with primary osteoporosis which is the other aim of your study.

RESULTS:

Tab.1: the total number of senior adults is 9, but then you have 5 female and 3 males so one patient is missing.

Fig.2: I would recommend to use a larger font size and to ameliorate the graphic aspect of the significance bars.

Line 209: The paragraph title is “3.4. Percentage of COP cells in Osteoporosis patients and evolution during treatment”, however you discuss only the COP in osteoporosis patients in this paragraph so you should remove the second part of the title which is discussed in the following paragraph.

Line 212-213: this sentence is twisted, please re-write it differently.

Fig.3: I would recommend to use a larger font size and to ameliorate the graphic aspect of the standard deviation bars of TPTD and Dmab at baseline and 12 months as they overlap. The caption is twisted, please re-write it differently.

Fig.4:  you used a different font with respect to the other figures, I would use the same font throughout the whole manuscript.

Tab.3: although p is not significant I would report the values instead of “NA”. I would also add the % changes which have more visual impact.

In general, you should always use the same number of decimal places when reporting results (e.g. line 213: either you should write 8.4 ± 7.0 vs. 3.5 ± 1.9, p= 0.027 if you want to use one decimal place or 8.40 ± 7.03 vs. 3.52 ± 1.94, p= 0.027 if you want to use two decimal places).

DISCUSSION: As showed in Fig.3, although the percentage of COP is similar at the end of the observation period, it appears that patients treated with TPTD have a higher AUC as compared with Dmab patients, therefore the overall amount of COP cells mobilized in these patients is higher. I would discuss this aspect and possible implications of that.

Round 2

Reviewer 2 Report

It's OK,and I accepted the response.